# Refractory Hypoxemia as a Trigger for Systemic Thrombolysis in Intermediate-High-Risk Pulmonary Embolism: A Case Report

**DOI:** 10.3390/reports8040253

**Published:** 2025-11-29

**Authors:** Ilias E. Dimeas, Panagiota Vairami, George E. Zakynthinos, Cormac McCarthy, Zoe Daniil

**Affiliations:** 1Department of Respiratory Medicine, University Hospital of Larissa, 41110 Larissa, Greece; 2School of Medicine, University College Dublin, D04 V1W8 Dublin, Ireland; 3Department of Respiratory Medicine, St. Vincent’s University Hospital, D04 T6F4 Dublin, Ireland; 4Department of Intensive Care, University Hospital of Larissa, 41110 Larissa, Greece; 53rd Department of Cardiology, “Sotiria” Chest Diseases Hospital, Medical School, National and Kapodistrian University of Athens, 11527 Athens, Greece

**Keywords:** pulmonary embolism, intermediate-high risk, systemic fibrinolysis, thrombolytic therapy, refractory hypoxemia, respiratory failure, multidisciplinary management, case report

## Abstract

**Background and Clinical Significance**: Intermediate-high-risk pulmonary embolism is characterized by right-ventricular dysfunction and positive cardiac biomarkers in the absence of hemodynamic instability. Current guidelines recommend anticoagulation with vigilant monitoring, and reserve systemic fibrinolysis for patients who deteriorate hemodynamically. However, some patients may experience physiologic decompensation manifested by refractory hypoxemia rather than hypotension, despite preserved systemic perfusion and normal lung parenchyma. In such cases, oxygenation failure reflects the severity of perfusion impairment and incipient right-ventricular-circulatory collapse. Whether this scenario justifies systemic fibrinolysis remains uncertain. **Case Presentation**: We present a 75-year-old man, five days after arthroscopic meniscus repair, presenting with acute dyspnea, tachycardia, and severe respiratory failure despite normal chest radiography. Laboratory findings revealed elevated troponin-I and brain natriuretic peptide, and echocardiography demonstrated marked right-ventricular dilation. Computed tomographic pulmonary angiography confirmed extensive bilateral central emboli with preserved lung parenchyma. Despite high-flow nasal oxygen at 100% fraction of inspired oxygen, respiratory failure worsened, necessitating intubation under lung-protective settings. With catheter-directed therapy unavailable and transfer unsafe, a multidisciplinary team administered staged systemic fibrinolysis with alteplase, pausing heparin during infusion. No bleeding or surgical complications occurred. Oxygenation and right-ventricular indices improved promptly. The patient was extubated on day 2, discharged from intensive care unit on day 7, and remained asymptomatic with normal echocardiography at 3 months. **Conclusions**: Refractory hypoxemia in intermediate-high-risk, normotensive pulmonary embolism, particularly when parenchymal disease and ventilator confounding are excluded, may represent an early form of circulatory decompensation warranting rescue reperfusion. In the absence of catheter-directed options and with acceptable bleeding risk, staged full-dose systemic fibrinolysis can be life-saving and physiologically justified. This case supports expanding the concept of “clinical deterioration” in intermediate-risk pulmonary embolism to include isolated, unexplained respiratory failure, highlighting the need for future trials to refine individualized reperfusion thresholds.

## 1. Introduction and Clinical Significance

Pulmonary embolism (PE) is a common and potentially life-threatening cardiovascular emergency, ranking third among acute cardiovascular diseases worldwide after myocardial infarction and stroke [1,2]. The clinical spectrum of PE ranges from asymptomatic or mild presentations to massive embolism with shock or cardiac arrest. Risk stratification is essential for guiding management, as mortality and complications differ substantially between high-, intermediate-, and low-risk categories [1,3].

According to the 2019 European Society of Cardiology (ESC) guidelines, patients with intermediate-high-risk PE are defined by the presence of both right ventricular (RV) dysfunction on imaging and positive cardiac biomarkers, but without hemodynamic instability [1]. These patients represent a challenging subgroup because, although normotensive at presentation, approximately 10–15% may experience early hemodynamic deterioration and require escalation of therapy [2,4]. The PEITHO trial demonstrated that systemic fibrinolysis with tenecteplase reduced hemodynamic collapse but increased major bleeding and intracranial hemorrhage, leading current guidelines to recommend anticoagulation and close monitoring as standard of care, reserving fibrinolysis for rescue therapy upon clinical deterioration [2,5].

However, the definition of “clinical deterioration” remains primarily focused on hemodynamic parameters, potentially overlooking cases of physiologic decompensation manifested as refractory hypoxemia with preserved blood pressure. Severe gas exchange impairment in PE is usually attributed to ventilation-perfusion mismatch caused by extensive perfusion defects, and may precede circulatory collapse [6,7]. Reports of intermediate-high-risk PE with severe respiratory failure despite normal lung parenchyma suggest that oxygenation failure alone may represent an early manifestation of impending RV failure and circulatory decompensation [8,9,10,11].

The management of such patients is particularly complex when catheter-directed therapy (CDT) or extracorporeal membrane oxygenation (ECMO) are unavailable, as delaying reperfusion in the face of escalating hypoxemia can be fatal. Recent expert position papers emphasize the need for individualized reperfusion strategies in these gray-zone scenarios, balancing physiological progression and bleeding risk [4,5,12,13]. Furthermore, case-based evidence supports the potential role of staged or reduced-dose systemic fibrinolysis as a safer alternative in carefully selected intermediate-high-risk patients [8,10,11,14].

Here, we present a case of intermediate-high-risk, normotensive PE complicated by refractory hypoxemia and persistent RV strain despite mechanical ventilation, successfully managed with staged full-dose systemic fibrinolysis following multidisciplinary team consensus. This case highlights the concept of respiratory failure as physiologic decompensation and proposes that oxygenation collapse, when parenchymal and ventilator causes are excluded, may warrant rescue reperfusion even in the absence of hypotension.

## 2. Case Presentation

A 75-year-old man presented to the emergency department with acute onset of dyspnea and pleuritic chest discomfort that had progressively worsened over several hours. He had undergone arthroscopic meniscus repair five days earlier without complications. His medical history included well-controlled hypertension but no prior venous thromboembolism, active malignancy, or cardiopulmonary disease. He was not on anticoagulation and had no history of bleeding disorders.

On arrival, he was tachypneic (respiratory rate 34 breaths/min, normal: 12–20), tachycardic (heart rate 118 beats/min, normal: 60–100), normotensive (blood pressure 114/72 mmHg), and afebrile. His oxygen saturation on room air was 72% (normal: ≥92%), which increased to 81% after initiation of high-flow nasal cannula oxygen therapy at 60 L/min with fraction of inspired oxygen (FiO_2_) 100%. Arterial blood gas analysis under these settings revealed a partial pressure of oxygen in arterial blood (PaO_2_) of 51 mmHg (normal: 80–100 mmHg; PaO_2_/FiO_2_ ratio = 51), partial pressure of carbon dioxide in arterial blood (PaCO_2_) 29 mmHg (normal: 35–45 mmHg), pH 7.45, and lactate 2.5 mmol/L (normal: <2 mmol/L), consistent with severe hypoxemic respiratory failure without metabolic alkalosis. Despite maximal non-invasive support, oxygenation remained critically low.

Physical examination revealed jugular venous distension and clear lung fields on auscultation without wheezes, crackles, or signs of consolidation. There was no lower-limb swelling or tenderness. Cardiac auscultation was normal, with no murmurs or gallops.

A supine chest radiograph demonstrated normal cardiac size and clear lung fields, without focal consolidation, pleural effusion, or pneumothorax (Figure 1).

No abnormalities were identified that could explain the profound hypoxemia. Electrocardiography revealed sinus tachycardia with an S_1_Q_3_T_3_ pattern and incomplete right bundle-branch block, consistent with right-heart strain.

Initial laboratory evaluation showed elevated high-sensitivity troponin-I 112 ng/L (reference range: <14 ng/L), B-type natriuretic peptide (BNP) 825 pg/mL (reference range: <100 pg/mL), and markedly increased D-dimer >20,000 ng/mL (reference range: <500 ng/mL), indicating myocardial injury and a high probability of pulmonary embolism.

Bedside transthoracic echocardiography demonstrated severe RV dilation with a D-shaped left ventricle and right atrial enlargement, consistent with acute pressure overload. The tricuspid annular plane systolic excursion (TAPSE) was 13 mm (reference range: >17 mm), confirming RV systolic dysfunction. There was no pericardial effusion, and the interventricular septum showed paradoxical motion, further supporting RV strain.

Given these findings, computed tomography pulmonary angiography was performed and revealed extensive bilateral central pulmonary emboli, involving both main pulmonary arteries and extending into the lobar and segmental branches (Figure 2). The lung parenchyma appeared entirely normal, without evidence of consolidation, ground-glass opacities, interstitial thickening, or atelectasis, confirming that the severe respiratory failure was not attributable to parenchymal disease, but rather reflected the extent of vascular occlusion and resulting perfusion impairment.

Despite high-flow nasal oxygen at maximal settings, the patient’s respiratory distress worsened, with progressive tachypnea and use of accessory muscles. He was therefore electively intubated for severe hypoxemic respiratory failure and placed on lung-protective mechanical ventilation (tidal volume 6 mL/kg predicted body weight, plateau pressure < 28 cmH_2_O, PEEP 8 cmH_2_O, FiO_2_ 100%).

He was then transferred to the intensive care unit for ongoing management. Under mechanical ventilation, arterial blood gas analysis showed PaO_2_ 62 mmHg (PaO_2_/FiO_2_ = 62) with stable hemodynamics (mean arterial pressure 82 mmHg, no vasopressor requirement). Repeat echocardiography demonstrated persistent RV dilation and dysfunction without improvement. The marked hypoxemia prompted a targeted assessment for intracardiac shunting. A transthoracic echocardiogram performed prior to his knee surgery showed no evidence of a patent foramen ovale (PFO) or intracardiac shunt. In addition, an agitated-saline contrast “bubble” study was performed in the ICU during the period of elevated right-sided pressures and was negative. Because increased right-atrial pressure in acute pulmonary embolism facilitates right-to-left bubble passage, a negative saline-contrast study under these conditions provides strong evidence against the presence of a PFO; therefore shunt was effectively excluded.

Given the ongoing refractory hypoxemia, extensive bilateral clot burden, and absence of alternative explanations for gas exchange failure, a multidisciplinary team, including respiratory medicine, cardiology, intensive care, hematology, and surgical representatives, convened urgently. CDT and ECMO were unavailable, and inter-hospital transfer was deemed unsafe due to the patient’s instability. Although recent orthopedic surgery represented a relative contraindication, surgical evaluation confirmed a low bleeding risk, and the consensus favored immediate reperfusion therapy.

Systemic fibrinolysis was initiated using a staged full-dose alteplase regimen: 50 mg infused over 2 h, followed by a second 50 mg dose after 24 h, with heparin temporarily withheld during infusion. No bleeding or surgical complications occurred. Within hours of the first infusion, oxygenation was stabilized and echocardiography showed partial RV recovery. After the second dose, further improvement in gas exchange and RV function was documented. The patient was successfully extubated on day 2, transferred out of the intensive care unit on day 7, and discharged home in stable condition on day 10. At 3-month follow-up, he remained asymptomatic while receiving a direct oral anticoagulant (rivaroxaban 20 mg once daily), with normal echocardiographic parameters and no evidence of pulmonary hypertension. After six months of anticoagulation, therapy was discontinued without recurrence, as the pulmonary embolism was attributed to a transient provoking factor, the recent arthroscopic knee surgery. A repeat transthoracic echocardiogram at that time again demonstrated normal right ventricular size, normal pulmonary pressures, and no evidence of pulmonary hypertension.

## 3. Discussion

Intermediate-high-risk pulmonary embolism represents a heterogeneous clinical entity characterized by right ventricular dysfunction and myocardial injury in the absence of systemic hypotension. According to the 2019 ESC Guidelines, anticoagulation remains the standard therapy for such patients, while systemic fibrinolysis is reserved for those who subsequently deteriorate hemodynamically [1]. This binary approach, however, fails to address the subset of patients who experience progressive physiological deterioration without hypotension. In this group, worsening gas exchange may reflect impending right ventricular failure and severe perfusion mismatch, despite preserved systemic blood pressure and normal lung parenchyma. Contemporary reviews emphasize that this “gray zone” of intermediate-high PE remains the most challenging for clinical decision-making and requires individualized, physiology-based management [15].

Disproportionate hypoxemia in acute pulmonary embolism may also raise the possibility of intracardiac shunting, particularly through a PFO, as shunt physiology can markedly worsen oxygenation in the setting of elevated right-sided pressures. Case-based evidence, including the report by Liew et al. [16] describing refractory hypoxemia from PFO-associated shunting in submassive PE, highlights this mechanism as an important diagnostic consideration. Although this phenomenon should be considered in similar presentations, intracardiac shunting was excluded in the present case, and the profound hypoxemia was instead attributable to severe perfusion impairment and RV-pulmonary vascular uncoupling

In acute pulmonary embolism, hypoxemia results mainly from increased dead space and ventilation-perfusion mismatch rather than alveolar flooding or atelectasis [1]. Thus, a profound reduction in the PaO_2_/FiO_2_ ratio may be an early indicator of extensive vascular obstruction and progressive RV strain. Recent data support incorporating oxygenation indices into PE risk assessment, as lower PaO_2_/FiO_2_ ratios correlate with adverse outcomes and early clinical deterioration [17]. In the present case, a low PaO_2_/FiO_2_ ratio before intubation indicated critical perfusion failure despite normotension, suggesting an imminent transition toward circulatory collapse. Similar cases of refractory hypoxemia preceding hemodynamic instability have been described in intermediate-high-risk PE [8,9], reinforcing the view that “clinical deterioration” should not be defined solely by systemic hypotension.

From a pathophysiological perspective, severe hypoxemia in PE reflects the combined impact of pulmonary vascular obstruction, increased RV afterload, and impaired oxygen delivery. Positive-pressure ventilation may exacerbate RV strain by increasing intrathoracic pressure and reducing venous return, yet in this case, respiratory failure preceded intubation and persisted despite protective settings, excluding ventilator-induced worsening. The constellation of RV dilation, elevated cardiac biomarkers, and critical hypoxemia constitutes a continuum toward RV ischemia and hemodynamic collapse [2,3,7]. Systemic fibrinolysis promptly reduces pulmonary vascular resistance, restores right-left ventricular coupling, and improves oxygenation, interrupting the downward spiral of RV failure [4,5,6,12].

The decision to proceed with systemic fibrinolysis was made after multidisciplinary consultation among respiratory medicine, cardiology, intensive care, hematology, and surgical teams. The patient fulfilled intermediate-high-risk criteria with imaging-confirmed RV dysfunction and persistent respiratory failure despite high-flow oxygen and mechanical ventilation. CDT and ECMO were unavailable, and inter-hospital transfer was considered unsafe due to ongoing decompensation. Although recent knee surgery represented a relative contraindication, surgical evaluation indicated minimal bleeding risk, and the benefits of reperfusion outweighed the potential harm. Given that current guidelines do not demonstrate an overall mortality benefit for thrombolysis in intermediate-high-risk PE and the bleeding risk is significantly increased, the decision in this case required explicit individualized risk-benefit analysis rather than extrapolation from general recommendations. The rise in high-sensitivity troponin and the marked elevation of NT-proBNP reflected significant acute cardiac strain rather than any chronic process. Taken together with persistent, worsening respiratory failure and imaging-proven, non-improving RV dilation, these findings indicated ongoing physiological deterioration despite preserved blood pressure. This biomarker pattern contributed to the individualized decision in this case to pursue rescue fibrinolysis in the absence of other available reperfusion options. A staged full-dose alteplase regimen (50 mg followed by 50 mg after 24 h) was selected, enabling early reassessment after partial lysis. This approach aligns with contemporary data supporting flexible dosing or slow-infusion regimens to balance efficacy and bleeding risk [11,14,18]. Although alteplase remains the standard fibrinolytic agent, data from tenecteplase studies have also demonstrated improved right ventricular function without excess major bleeding, reinforcing the broader concept of early, individualized reperfusion in selected intermediate-high-risk patients [19]. Registry evidence suggests that individualized dosing strategies can optimize outcomes across intermediate-high-risk phenotypes [20].

The PEITHO trial demonstrated that systemic fibrinolysis prevents hemodynamic collapse but increases major bleeding and stroke [2]. Yet most trial participants were not severely hypoxemic, and mechanical ventilation was rare. Subsequent meta-analyses reinforced that while thrombolysis reduces mortality, the overall benefit depends on individualized risk-benefit evaluation [21,22]. Our case extends this evidence by suggesting that refractory hypoxemia in the absence of parenchymal disease may signal a physiologic form of circulatory compromise deserving of early reperfusion. Such oxygenation failure represents a pre-shock phenotype, in which delaying reperfusion carries a substantial risk of abrupt hemodynamic collapse despite preserved systemic blood pressure. Observational studies have shown that half-dose or slow-infusion regimens can shorten the duration of hypoxemia and accelerate RV recovery [11,14,23], while registry data indicate that rapid oxygenation decline and biomarker elevation identify the subgroup most likely to deteriorate [7,24]. These findings support a more dynamic approach to treatment selection, integrating both respiratory and hemodynamic trajectories.

This experience underscores the importance of integrating respiratory failure into the risk continuum of PE. In settings where CDT or ECMO are not available, systemic fibrinolysis, particularly when applied in a staged or slow-infusion protocol, remains a viable rescue option when bleeding risk is acceptable [4,5,14,18,20]. Furthermore, the case highlights the limitations of current guideline definitions, which equate clinical deterioration exclusively with hypotension [1]. Recognizing refractory hypoxemia as an equivalent marker of decompensation could refine therapeutic decision-making and ensure earlier reperfusion in otherwise normotensive patients.

Certain limitations must be acknowledged. This is a single case, and conclusions regarding efficacy cannot be generalized. No invasive hemodynamic measurements or continuous cardiac output data were available during the acute phase, limiting precise physiologic interpretation. Small changes in driving pressure could have influenced RV performance despite lung-protective settings. Moreover, as ECMO and CDT were unavailable, management decisions reflected pragmatic, real-world practice rather than protocolized escalation. Ventilation-perfusion scintigraphy was not feasible in this critically ill, mechanically ventilated patient; however, the extensive central clot burden and normal lung parenchyma on CT pulmonary angiography made additional perfusion imaging unlikely to change acute management. Nonetheless, the temporal association between thrombolysis and rapid oxygenation and RV recovery provides a strong physiological rationale for the observed benefit. However, the use of systemic fibrinolysis in normotensive intermediate-high-risk PE remains controversial, and decisions must be individualized with careful multidisciplinary evaluation of bleeding risk.

In summary, this case demonstrates that refractory hypoxemia in intermediate-high-risk pulmonary embolism may represent an early stage of circulatory decompensation even in normotensive patients. When catheter-based therapies are unavailable, timely systemic fibrinolysis, guided by multidisciplinary consensus and careful risk assessment, can restore oxygenation and RV function without major bleeding. These observations support expanding the definition of “clinical deterioration” to include isolated, unexplained respiratory failure, emphasizing the need for future studies to establish standardized, physiology-based reperfusion criteria.

## 4. Conclusions

This case highlights the potential role of refractory hypoxemia as a manifestation of physiological, rather than purely hemodynamic, decompensation in intermediate-high-risk pulmonary embolism. In patients presenting with severe oxygenation failure, preserved blood pressure, and normal pulmonary parenchyma, respiratory collapse may signify early right ventricular failure and imminent circulatory compromise. When catheter-directed reperfusion or extracorporeal support are unavailable, timely systemic fibrinolysis, preferably administered in a staged or slow-infusion protocol, can be a rational and life-saving intervention if bleeding risk is acceptable and multidisciplinary agreement is achieved. Although this single case cannot redefine treatment thresholds, it underscores the need for future prospective studies to evaluate whether refractory hypoxemia should be incorporated into the definition of clinical deterioration in pulmonary embolism. Recognizing oxygenation failure as a physiological equivalent of impending shock may refine individualized decision-making and support earlier, targeted reperfusion strategies in carefully selected patients.

## Figures and Tables

**Figure 1 reports-08-00253-f001:**
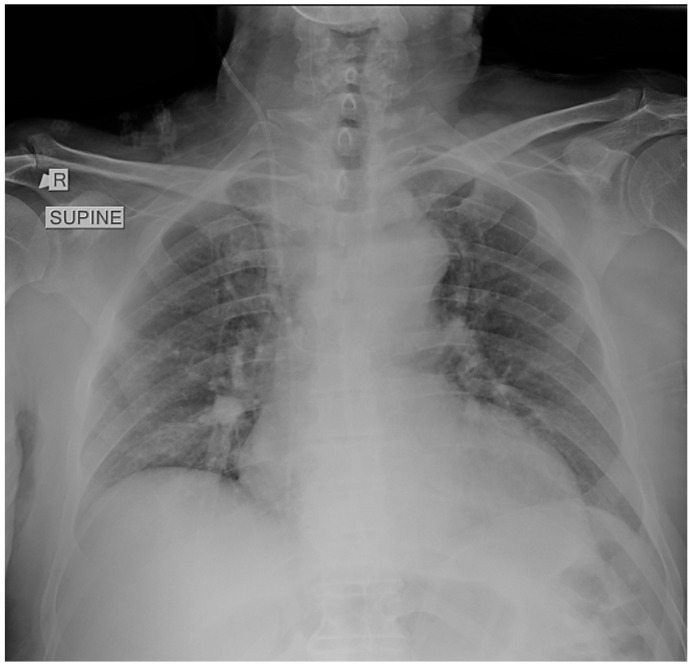
Supine chest radiograph obtained in the emergency department showing no focal consolidation, pleural effusion, or pneumothorax. The lung fields and costophrenic angles are clear, and cardiac size is within normal limits. No radiographic abnormalities are identified to explain the patient’s profound respiratory distress.

**Figure 2 reports-08-00253-f002:**
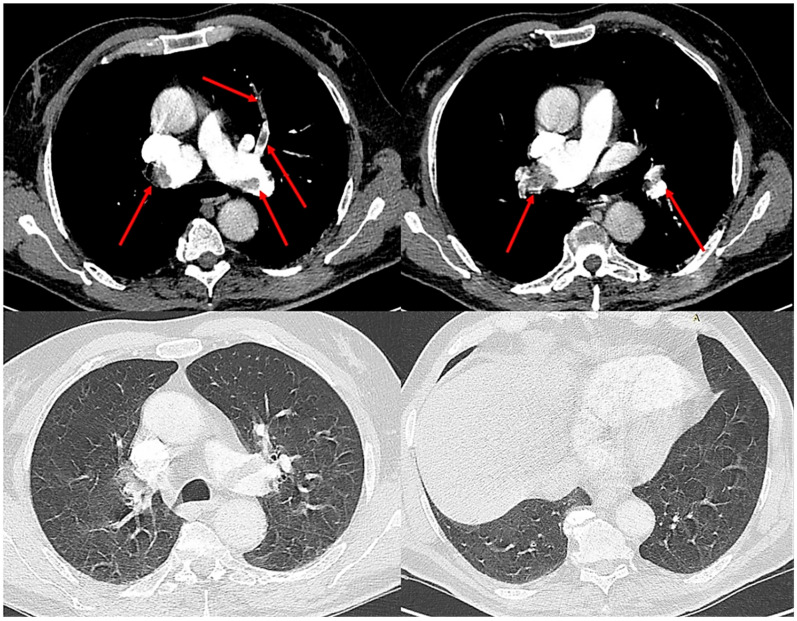
Computed tomography pulmonary angiography demonstrating extensive bilateral central pulmonary emboli (red arrows) involving both main pulmonary arteries and segmental branches. The lung parenchyma appears normal, without evidence of consolidation, ground-glass opacities, or atelectasis, confirming that the severe hypoxemia was not attributable to parenchymal disease.

## Data Availability

The original contributions presented in this study are included in the article. Further inquiries can be directed to the corresponding author.

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
