# Peer review of "Refractory Hypoxemia as a Trigger for Systemic Thrombolysis in Intermediate-High-Risk Pulmonary Embolism: A Case Report"

_reports, 2025, doi:10.3390/reports8040253_

Round 1

Reviewer 1 Report

Comments and Suggestions for Authors

While this case report is interesting, several important issues should be highlighted:

There is a well-written case report from 2018 on submissive pulmonary embolism with refractory hypoxemia (Perm J. 2018 Mar 2;22:17-061. doi: 10.7812/TPP/17-061). The authors reviewed similar cases and emphasized the importance of considering patent foramen ovale (PFO) and intracardiac shunts. Although there is no information about transesophageal echocardiography (TEE) or echo contrast for the current patient, the possibility remains likely.

Current guidelines do not suggest an overall mortality benefit from thrombolytic therapy in cases of submissive pulmonary embolism; however, the bleeding risk increases significantly. Therefore, the risk-benefit ratio of using this approach in selected patients should be carefully addressed.

Given these considerations, the discussion and limitations section of the report requires appropriate revision.

Author Response

Dear Reviewer,

We have carefully revised the manuscript following your thoughtful and constructive comments. Below, we address each point and outline the specific modifications made.

1. Comment regarding the need to consider PFO/intracardiac shunting

Thank you for highlighting the 2018 report describing refractory hypoxemia in submassive PE associated with a patent foramen ovale. This is indeed an important differential diagnosis in patients who appear more hypoxemic than expected from imaging findings. In the revised version, we added a short discussion referring to this report and clarifying the relevance of shunt physiology. We also clarified in the Case Presentation that our patient underwent two separate assessments, a pre-operative transthoracic echocardiogram and an ICU saline-contrast (“bubble”) study during elevated right-sided pressures, both of which were negative. These findings effectively excluded a PFO in this case. The Discussion now acknowledges this mechanism in similar presentations and states explicitly that it was ruled out here.

2. Comment regarding guideline evidence, mortality benefit, and bleeding risk of thrombolysis

We fully agree that current guidelines do not demonstrate an overall mortality benefit for thrombolysis in intermediate-high-risk patients and that bleeding risk is a major concern. This was not sufficiently emphasized in the original version. We have now expanded the Discussion to present this point more clearly and to explain the reasoning behind the clinical decision in this specific case. In the revised manuscript, we describe the risk–benefit considerations more transparently: the recent knee surgery was reviewed by the orthopedic team and deemed low risk for bleeding; CDT and ECMO were unavailable; transfer was considered unsafe; and the patient’s severe hypoxemia, despite normal lung parenchyma, represented physiological deterioration toward circulatory compromise rather than isolated respiratory failure. We added a sentence noting that the decision required individualized risk–benefit assessment rather than straightforward application of guideline thresholds.

3. Comment regarding the need for revision of the Discussion and Limitations

Both the Discussion and Limitations sections have been revised in line with your feedback.
In the Discussion, we expanded the explanation of why refractory hypoxemia can represent early circulatory decompensation even in normotensive patients, and why this scenario may justify rescue reperfusion when other options are unavailable. We also added a sentence clarifying that this pattern represents a pre-shock phenotype, in which delaying reperfusion carries a real risk of sudden hemodynamic collapse.

In the Limitations section, we added an explicit acknowledgment that the use of systemic fibrinolysis in normotensive intermediate-high-risk PE remains controversial, and that decisions must be individualized with careful multidisciplinary evaluation. This important point is now clearly reflected.

We hope that the revised manuscript addresses all of your concerns and improves the clarity and value of the report. Thank you again for your time and constructive input.

Kind regards,
Corresponding Author

Reviewer 2 Report

Comments and Suggestions for Authors

The case raises hystorical pitfalls in the management of selected type of PE.

authors clarify that fibrinolysis is controversial but still use mainly in haemodynamic PE so i suggest to better clarify why they considered a different clinical method to select fibrinolysis as therapeutic option.

furthermore, the persistent hypoxia may be related to several other causes mainly the ventilatory / perfusion mismatch that is better evaluated with scintigraphy that is not reported and should be recognized as study limitation and also to increased artery pulmonary pressure per se or for other chronic illness . This aspect of pulmonary hypertension as keyway for the hypoxia and the choice of pharmacological fibrinolysis may be better described in the text in particular in discussion and associated to levels and roles of cardiac biomarkers as TPN and NT pro BNP.

Author Response

Dear Reviewer,

We have revised the text accordingly. Below, we address each point in turn and summarize the changes made.

1. Clarification regarding the choice of fibrinolysis

We appreciate your comment regarding the need for a clearer explanation of why systemic fibrinolysis was selected in this normotensive, intermediate-high-risk case. In the revised Discussion, we expanded the reasoning to emphasize the individualized nature of this decision. Specifically, we clarified that the patient’s biomarker profile played an important role: the rise in high-sensitivity troponin indicated acute myocardial injury, while the marked elevation of NT-proBNP reflected substantial right-ventricular wall stress. When considered together with persistent severe hypoxemia, normal lung parenchyma on CT, and imaging-proven RV dilation without improvement, these biomarkers supported the conclusion that the patient was physiologically deteriorating despite preserved systemic blood pressure. We have added a short paragraph describing how this biomarker pattern contributed to the individualized decision to proceed with rescue fibrinolysis, especially in the absence of other available reperfusion strategies such as CDT or ECMO.

2. Comment regarding other possible causes of persistent hypoxemia

Thank you for raising the question of alternative explanations for the degree of hypoxemia observed. We have clarified this point in our response while keeping the manuscript focused and clinically coherent.

Regarding the possibility of other causes of persistent hypoxemia, we note that these were also evaluated. The patient had a completely normal transthoracic echocardiogram shortly before the event, with no evidence of pulmonary hypertension or other chronic cardiopulmonary disease. A repeat echocardiogram at six months again demonstrated normal pulmonary pressures and right-ventricular size, confirming that no pre-existing or chronic pulmonary vascular condition was present. In addition, the CT scan performed at presentation showed normal lung parenchyma, making chronic V/Q abnormalities or alternative respiratory pathology unlikely. Taken together, no other medical condition or chronic illness was identified that could explain the degree of hypoxemia observed. Although ventilation-perfusion scintigraphy could theoretically provide additional information about perfusion defects, it was not feasible in this ICU patient, and the extensive acute clot burden demonstrated on CT pulmonary angiography made additional perfusion imaging unlikely to change immediate management. However, a clarification was added in the limitations as per your suggestion.

We hope these clarifications address your concerns and improve the balance and clarity of the manuscript. Thank you again for your thoughtful and constructive input.

Kind regards,
Corresponding Author

Round 2

Reviewer 1 Report

Comments and Suggestions for Authors

The authors addressed the comments well.